# The Effect of Social Isolation on Physical Activity during the COVID-19 Pandemic in France

**DOI:** 10.3390/ijerph18105070

**Published:** 2021-05-11

**Authors:** Alessandro Porrovecchio, Pedro R. Olivares, Philippe Masson, Thierry Pezé, Linda Lombi

**Affiliations:** 1Univ. Littoral Côte d’Opale, Univ. Lille, Univ. Artois ULR 7369—URePSSS—Unité de Recherche Pluridiscipli-Naire Sport Santé Société, 59140 Dunkerque, France; thierry.peze@univ-littoral.fr; 2Faculty of Sport Sciences, Universidad de Huelva, 21007 Huelva, Spain; pedro.olivares@ddi.uhu.es; 3Institute of Physical Activity and Health, Universidad Autonoma de Chile, Talca 3460000, Chile; 4Univ. Lille, Univ. Littoral Côte d’Opale, Univ. Artois—ULR 7369—URePSSS—Unité de Recherche Pluridisciplinaire Sport Santé Société, 59000 Lille, France; philippe.masson@univ-lille.fr; 5Department of Sociology, Università Cattolica del Sacro Cuore, 20125 Milan, Italy; linda.lombi@unicatt.it

**Keywords:** lifestyles, social distancing, lockdown, sedentarity, education, covid-19

## Abstract

The objective of this cross-sectional study is to analyze the changes in physical activity (PA) practice of a sample of 2099 French adults, mostly females, who answered an online questionnaire during the first COVID-19 lockdown (March–May 2020). A descriptive analysis of participants was performed using relative frequencies. Chi-squared tests were performed to compare the responses of selected variables. Multinomial logistic regressions were performed to compare the variations of PA with all the variables identified. The age of participants ranged from 18 to 88. Among people who practiced PAs before the first lockdown, the probability to keep practicing PAs is higher among those with a lower level of education, among housewives and retirees and among those who lived in cities of 10,000–19,999 inhabitants. For those who did not practice PAs before the social distancing, the probability of starting to practice is greater in those with a lower level of education and for those who suffered from a chronic disease. Our results place the emphasis on the complexity and multifactoriality of the changes that emerged during the first lockdown. The “education” factor emerges, as a significant determinant of PA that should certainly be explored further.

## 1. Introduction

Since its first appearance in Wuhan (China), around mid-December 2019, the coronavirus pandemic has started to spread across the world. The steady increase in daily deaths and confirmed cases since the beginning of 2020 has prompted the governments of affected countries to adopt protection strategies, relying mainly on social distancing and other containment actions. To combat the epidemic, a so-called lockdown was put in place in France from 17 March to 11 May 2020: most workplaces and public places, including schools, shops, bars and restaurants, were closed or made accessible in a limited way. The daily lives of millions of French people have suddenly been transformed, leading to a significant change in lifestyles, family relationships and work routines, with significant consequences on their quality of life and psychological well-being, in the short, medium and long term [1].

In these extraordinary times the coronavirus crisis was been enormously covered and discussed in both the media and academia. From early on, this content also focused on the implications of the virus for sport, exercise and physical activity (PA) [2]. However, less is known about the consequences on the practice of PA and as concerns the changes in lifestyle in the short, medium and long term. In this paper, we analyze the impact of social distancing on the practice of PA of a sample of French adults who answered an online questionnaire.

Loneliness and social isolation are usually associated with poor mental and physical health and pose an important risk factor as regards the probability of experiencing the most common mental disorders (e.g., anxiety and depression) [3]. For example, sociability [4] and also non-sedentary lifestyles are usually associated with reduced overall mortality, an increase in life expectancy and a greater likelihood of living an old age in good conditions [5]. Unhealthy lifestyles and problematic behaviors are risk factors for physical [6] and mental health [7]. Strong evidence associates physical inactivity and sedentary behaviour with an increased risk of chronic diseases, which are the leading causes of death worldwide [8].

Scientific evidence leaves no doubt about the need to lead an active physical life to develop and protect one’s overall health at all ages [9]. According to the WHO [10], today, physical inactivity and a sedentary lifestyle are the fourth leading cause of death worldwide and are a major public health problem. Furthermore, sedentary lifestyles lead to physiological disorders, which in turn generate significant costs in terms of health expenditure [11]. PA promotion to prevent the pandemic spread of diseases linked to physical inactivity and sedentary lifestyles, and to improve populations’ health, has been for decades a core objective of health strategies and policies globally [12].

The coronavirus crisis radically changed this assumption. The mandated restrictions on PA, which widely affected those related to work, commuting, sport and exercise, disrupted the PA routine of millions of people and generated a contradictory situation. On the one hand, most kinds of PAs were perceived as risky behaviours, being a way to spread COVID-19. On the other hand, while taking precautions, PA remains an important tool to maintain the population healthy despite the lockdown [13,14,15]. Previous pandemic crises, as for example that caused by the severe acute respiratory syndrome (SARS), caused serious public health consequences, not only linked to the viral infection per se. Indirect impacts on communities’ health haven’t been assessed systematically, although early studies on the effects of quarantine and lockdown as protective measures show its negative psychological effects on the population [16]. Previous studies related to the SARS pandemic show that the community in Hong Kong responded by adopting healthier behaviours [17]. However, some authors hypothesize that this health crisis has the potential to further impact and accelerate the physical inactivity and sedentary behaviour pandemic we have been confronted with, and failing to address, for a number of years [18], and all the risks that follow from it [19,20].

Despite the scientific consensus and the deployment of incentive policies to promote PA and engage people in more active lifestyles, the phenomena of sedentary condition and inactivity are increasing, especially in high-income countries [21]. Many scholars engaged in public health’s analyzes of the present pandemic situation stress that both modifiable lifestyle factors like diet and PA [22] and mental health issues [23] should not be marginalized from policy makers’ considerations.

Studies from China, the center of the first wave of the epidemic, showed how nearly 60% of Chinese adults had inadequate physical activity (95% CI 56.6%–58.3%), which was more than twice the global prevalence (27.5%, 25.0–32.2%), during the early days of the novel epidemic [24]. Starting from this background, in this paper we focus on the changes in lifestyles of a sample of French adults during the first lockdown, in the months between March and May 2020, with an emphasis on PA. In particular, we test the following hypotheses: (1) the pandemic had an impact on the frequency of physical activity; (2) socio-demographic characteristics and health conditions influence the likelihood of physical activity or not during the breakdown.

## 2. Materials and Methods

### 2.1. Design

The results presented here are part of “Pandemic Emergency in Social Perspective. Evidence from a large Web-survey research”, an international exploratory research that studied the social and psychological impact of the physical distancing measures imposed by the COVID-19 pandemic in six European countries (Italy, United Kingdom, Sweden, Czech Republic, Poland and France). The online survey included a total of 31 questions that covered the following areas: (a) Demographic information (gender, age, marital status, employment status, family, number and age of children living at home, living conditions, residential area); (b) COVID-19 experience (safety and precautionary measures adopted to reduce the risk of contagion, social relationship during the quarantine, health emergency duration expectations, post-COVID scenario prevision, personal testing and outcome for COVID-19, loss of relatives or friends due to COVID-19); (c) COVID-19 and media source information (level of information perceived, information sources and channels); (d) COVID-19 risk perception (fear of getting sick, general and personal concern regarding the virus); (e) Lifestyle behaviours (diet, alcohol, and tobacco consumption during the pandemic, physical activity); (f) Perceived Stress: the 8-item version of Personal Health Questionnaire Depression Scale (PHQ-8), scored basing on Kroenke et al. [25]; (g) Health condition (chronic health conditions, general health status, daily activity abilities).

The survey was administered using the Qualtrics web survey platform. Data collection occurred between March and June 2020. The respondents were recruited through a “snowball” non-probabilistic sampling strategy, through the Facebook platform. In particular, we shared the link to our web survey in Facebook groups which were dedicated to the COVID-19 pandemic. The inclusion criterion was the age of majority. The study was approved by the Ethics Committee of the Policlinico Gemelli, Catholic University of the Sacred Heart of Rome (Prot. N. 00255223/20).

### 2.2. Statistical Analysis

A descriptive analysis of participants was performed using relative frequencies. Chi-squared tests were performed to compare the responses in the selected variables based on gender and PA practice during the period of social distancing. In order to calculate the odds ratio (OR) and its 95% confidence interval of the variables in which there were statistical significance based on PA practice during the period of social distancing, a multinomial logistic regression model was done using all these variables. Additionally, two other multinomial logistic regression models were done using the same variables in order to calculate the OR for the changes from “no sport practice before social distancing” to “practice during this period”, and from “sport practice before social distance” to “no practice during this period” as dependent variables, both of them dichotomous variables. All data were analyzed using SPSS software and statistical significance was considered as *p* < 0.05.

## 3. Results

We received 2410 answers in France. After excluding people who did not answer all questions, data of a total of 2099 participants, 81.6% of which were females, were considered for the analysis. The age of participants ranged from 18 to 88 with a mean age of 41.1 ±12.7. Table 1 shows the socio-demographic characteristics of participants as well as responses to health and PA’s practice questions by sex. Most participants were in a relationship, cohabitant or married (65.4%), with a university degree (62%), workers (68.4%) and lived in a city with less than 100.000 inhabitants (72.4%). As for people living in the same household during the social distancing period, only 15.5 % lived alone and 37.4% lived without children all or some of the time. Regarding the health related questions, only 5.2% reported bad or very bad health, 30.6% suffered chronic diseases, 10.5% reported serious limitations in their daily activities and 27.6% scored as “moderate”, “moderate severe” or “severe depression” in the Personal Health Questionnaire Depression Scale (PHQ-8) [26]. Concerning PA’s practice, 54.1% reported they regularly practiced sports before confinement and 57.3% said they practiced a PA during the period of social distancing (25.9% of them started to practice PA during this period). Analyzing changes in PA practice during the period of social distancing, 34.3% of people who did not practice any PA before this period started to, and 22.2% of those who practiced PA before this period stopped.

Focusing on PAs’ practice during the period of social distancing, we can find the answers on Table 2 (“No practice”; “Practice with the same frequency as before”; “Practice, although less frequently”; “Started to practice in the social distancing period”). These answers are based on the socio-demographic and health-related characteristics of participants. We found statistically significant differences based on all variables except for “marital status” (*p* = 0.059) and “number of people who live together” (*p* = 0.249).

Table 3 shows the results of the multivariate regression for the PAs’ practice during the social distancing period, using as reference category “not practiced AP during the social distancing period”, and Table 4 shows the results of the multivariate regression for the PA’s practice changes during the social distancing period. Among people who practiced PAs before the social distancing period, the probability to keep practicing PAs is higher among those with a lower level of education (1.96 times), among housewives and retirees (2.94 and 2.86 times respectively) and among those who live in cities of 10,000–19,999 inhabitants. Concerning those who did not practice PAs before the period of social distancing, the probability of starting to practice is greater in those with a lower level of education (3.12 and 2.22 for “lower secondary school or less” and “Diploma/upper secondary school” respectively) as well as for those who suffer from a chronic disease (1.51 times).

## 4. Discussion

As with many other recent research that analyzed lifestyle changes during the social distancing period, our findings are complex and somewhat ambiguous (i.e., as for alcohol consumption [27]). Some of our results were partially unexpected.

First, some aspects that we thought might have an effect on PA don’t seem so significant. This is the case of the age category, which does not seem to have a significant effect on its variation during the first lockdown. Consistently with other research relating to the effect of age on the quantity and quality of physical activity [28], we would have expected a less ambiguous correlation during the social distancing period too and a more important “age effect”.

Another unexpected result was related to those who did not practice PAs before the first lockdown: those who suffered from a chronic disease had 1.51 times more probability of starting to practice. This is a population at risk of a sedentary lifestyle which, overall, is not always easy to involve in PA. In particular, a recent study on lifestyle adherence in stay-at-home patients with chronic coronary syndromes found that almost half (45%) of participants reported a reduction in physical activity during first French lockdown [29]. 

Other aspects that we took for granted have been confirmed by data analysis: for example, concerning the psychological aspects that we have considered using the PHQ-8 questionnaire, as other studies point out too [30] the fact of being in an healthy or “normal” mental health condition (we refer to the PHQ questionnaire’s categories “no depression” or “minimal depression”), allows to give continuity over time to the practice of physical activity, and we can also assume that it helps to maintain a healthy lifestyle. Conversely, many cross-sectional studies have reported that depressed patients are more sedentary [31]. However, this association may be bidirectional: problematic mental health may lead to decreased levels of exercise due to low motivation and energy and decreased exercise may be a risk factor for depression [30].

If we focus on healthier conducts, our data show that having a lower educational level and living in a small town (or—we suppose basing on the classification proposed by Jousseaume and Talandier in 2016 [32] concerning the French context—in a rural area town, but our questionnaire doesn’t allow us to confirm this aspect, unfortunately) are positive factors of influence (determinants) with respect to the fact of continuing practicing PAs during the first lockdown, or starting practicing a PA if they never did. An hypothesis to explain why people who leave in a small town practice more PA could be that they went out even if there was a ban on going out (in a rural area there is less control): our analysis show a strong correlation between the fact of living in one of these towns and the tendency to practice PA, and thus confirms this hypothesis. Our data does not allow us to point out the difference between those who have enough space to practice in their living environment and those who do not. We can assume that the lockdown experience changed something in the perception and future design of living spaces. A survey carried out on a sample of 1056 people by Opinion Way for Artémis Courtage in June 2020 highlights the new appeal of housing with outdoor space: 10% of participants declared they wanted to move in a rural area; 29% of twenty-five-to-thirty-fours wanted a garden and 23% a terrace [33].

As for the educational level and the place of residence, we note that there is also a strong correlation with regard to whether or not to practice sport before the first lockdown (Table 2): again, there is a more important number of people “not practicing sport” among those with a “lower secondary school” or a “diploma/upper secondary school” educational level, living in small towns. Furthermore, those having a “lower secondary school” educational level are the ones having a positive balance between those who practiced less and those who started. But looking more deeply, those who have a low level of education are also those who have shown a greater variability in the practice of physical activity between before and during the first lockdown (Table 4).

Finally, a last interesting aspect related to healthy practices: we note an interesting continuity in the practice of PA among retirees who practiced even before the first lockdown. This is certainly an interesting indicator of a healthy lifestyle that tries to remain so even in situations of crisis. Furthermore, recent studies have clearly demonstrated the mental health benefits of physical activity in adults [34] and older adults [35] during lockdown. In particular, in a recently published study, Faulkner et al. [34] show that participants who reported a negative change in exercise behaviour during the initial COVID-19 restrictions demonstrated poorer mental health and well-being compared to those demonstrating either a positive-or no change in their exercise behaviour.

Our data, in fact, show overall a positive increase in the amount of PA practice in our sample, which is an indicator of improvement lifestyles during the first lockdown. This result is in contrast to a Spanish research in which a decrease in PA was found [36].

## 5. Conclusions

Certainly, our results place the emphasis, once again and in accordance with what emerged in the literature, on the complexity and multifactoriality of the changes that emerged during the first lockdown, in relation to the various profiles of respondents. In this complexity, the “education” factor emerges, as a significant determinant of PA that certainly has to be explored further. In order to understand these results, it will be necessary to integrate other variables, such as the motivation, and to explore the subjective dimension of the experience of social distancing, in relation to the practice of PAs.

From the data in our possession there does not seem to be a main influencing factor (determinant) of PA during the first lockdown. This leads us to open new avenues for reflection: from a perspective of prevention, accompaniment of change and/or intervention, it is important to identify the main factors of influence (or determinants) of PA. These are some factors that we have chosen not to analyze, given the descriptive and exploratory nature of our work, but which emerge between the lines of our results, when, analyzing the impact of the lockdown, we can observe the differences between some categories and the role of the educational level emerges ambiguously.

The analysis of these factors could bring out the social inequalities at the basis of the changes that we have observed and could therefore provide the tools to be able to build adapted health and PA policies if some new lockdowns occur.

## 6. Limits of the Study

The respondents were recruited through a “snowball” non-probabilistic sampling strategy, through the Facebook platform. This means that the people who responded to our questionnaire were also the most active on social media, especially on Facebook. Although non-probabilistic sampling strategies do not allow to obtain representative samples of the entire population, in accordance with Brickman Bhutta [37], the administration of an online survey through the use of social networks offers new opportunities for scholars to collect data faster, at lower cost and with less need for assistance for responding compared to what could be possible through traditional data collection methods: for these reasons, “*Facebook may be a useful tool for exploratory work and for rapid pretesting of surveys destined for dissemination* via *traditional method*” ([37], p. 59). Additionally, the cross-sectional design limits the ability to draw on causal associations.

## 7. Key Points

Among those who practiced PA before the social distancing period, the probability of keeping practicing PA was higher among those with a lower level of education, housewives, retirees, and those who lived in small cities.As for those who did not practice PA before the social distancing, the probability of starting to practice is greater in those with a lower level of education and for those who suffered from a chronic disease.Our results place the emphasis on the complexity of the changes that emerged during the first lockdown.In this complexity, the “education” factor emerges as a significant determinant of PA that certainly should be explored further.

## Figures and Tables

**Table 1 ijerph-18-05070-t001:** Sociodemographic characteristics of participants (*n* = 2099, 387 males and 1712 females).

Variable	Male	Female	All	*p*
Age group				0.003
18–24	15.0%	9.6%	10.6%	
25–34	22.7%	22.7%	22.7%	
35–44	22.0%	27.9%	26.8%	
45–54	20.7%	24.0%	23.4%	
55–64	14.7%	12.6%	13.0%	
65+	4.9%	3.2%	3.5%	
Marital status				
Single	27.6%	20.4%	21.8%	0.001
In a relationship and cohabitant	31.3%	35.1%	34.4%	
Married	32.6%	30.7%	31.0%	
Separated, divorced or widow	8.5%	13.8%	12.8%	
Educational level				
No education or Primary school	0.3%	0.4%	0.4%	0.018
Lower secondary school	7.8%	9.7%	9.3%	
Diploma/upper secondary school	15.8%	16.0%	15.9%	
Degree	58.9%	62.7%	62.0%	
Master, PhD or post-degree	17.3%	11.2%	12.3%	
Professional status				
Worker	66.7%	68.8%	68.4%	0.001
I am looking for a new job	7.5%	6.8%	6.9%	
Housewife	0.5%	3.9%	3.3%	
Student	12.1%	9.2%	9.7%	
Retired	8.5%	5.6%	6.1%	
Other conditions	4.7%	5.7%	5.5%	
City inhabitants				
500,000 or more	16.1%	9.6%	10.8%	0.007
250,000–499,999	5.2%	5.9%	5.8%	
100,000–249,999	12.2%	10.7%	11.0%	
20,000–99,999	23.7%	25.1%	24.8%	
10,000–19,999	9.6%	11.4%	11.0%	
Less than 10,000	33.1%	37.4%	36.6%	
How many people currently live in the house where you are spending your social distancing period or most of your time?				
I live alone	19.6%	14.5%	15.5%	0.054
2 persons	31.5%	30.7%	30.8%	
3 persons	19.1%	22.3%	21.7%	
4 or more persons	29.7%	32.5%	32.0%	
How many children do you have living with you all or some of the time				
None	50.5%	34.4%	37.4%	0.000
One	12.7%	16.6%	15.9%	
Two	23.6%	32.6%	30.9%	
Three or more	13.2%	16.4%	15.8%	
Generally, how is your health?				0.235
Very good/good	76.8%	72.5%	73.3%	
Nor good or bad	18.5%	22.2%	21.5%	
Bad/very bad	4.7%	5.3%	5.2%	
Are you suffering from chronic diseases or long-lasting health problems?				0.145
Yes	27.4%	31.3%	30.6%	
No	72.6%	68.7%	69.4%	
Due to health problems, do you have any limitations in your daily activities?				0.303
Serious limitations	9.4%	10.8%	10.5%	
Not serious limitations	16.0%	18.7%	18.2%	
No limitations	74.6%	70.6%	71.3%	
Depression level (PHQ-8)				0.000
No depression	10.9%	7.2%	7.9%	
Minimal	40.5%	29.3%	31.4%	
Mild	27.7%	34.4%	33.2%	
Moderate	12.0%	17.5%	16.5%	
Moderate severe or severe	8.8%	11.6%	11.1%	
Before social distancing, did you regularly practice PA?				0.163
Yes	57.3%	53.3%	54.1%	
No	42.7%	46.7%	45.9%	
During the period of social distancing, are you practicing PA?				0.347
No	39.1%	42.9%	42.2%	
Yes, with the same frequency as before	21.8%	18.9%	19.4%	
Yes, although less frequently	25.1%	23.1%	23.4%	
Yes, I’ve started to do it since I’ve been in social distancing period	14.0%	15.2%	15.0%	

Values in percentages. *p*: *p* of Chi-squared test.

**Table 2 ijerph-18-05070-t002:** Responses to During the period of social distancing, are you practicing PA? (*n* = 2099, 387 males and 1712 females).

Variable	No Practice	Practice with the Same Frequency as Before	Practice, although Less Frequently	Yes, Started to Practice in Social Distancing Period	*p*
Age category					0.000
18–24	36.8%	18.2%	18.2%	26.8%	
25–34	41.4%	21.1%	19.4%	18.1%	
35–44	42.5%	16.5%	26.3%	14.7%	
45–54	44.9%	21.0%	23.3%	10.8%	
55–64	43.6%	20.5%	24.5%	11.4%	
65+	37.8%	18.9%	40.5%	2.7%	
Marital status					
Single	36.7%	21.1%	24.4%	17.8%	0.059
In a relationship and cohabitant	42.2%	20.1%	21.6%	16.1%	
Married	43.4%	18.4%	25.5%	12.8%	
Separated, divorced or widow	48.3%	17.1%	21.9%	12.6%	
Educational level					
No education or Primary school	100.0%				0.000
Lower secondary school	61.2%	15.8%	11.2%	11.7%	
Diploma/upper secondary school	49.4%	16.2%	18.3%	16.2%	
Degree	38.5%	19.9%	26.0%	15.6%	
Master, PhD or post-degree	35.2%	24.2%	27.3%	13.3%	
Professional status					
Worker	41.5%	20.2%	23.8%	14.5%	0.000
I am looking for a new job	45.5%	21.4%	17.2%	15.9%	
Housewife	56.5%	13.0%	20.3%	10.1%	
Student	37.1%	16.3%	18.8%	27.7%	
Retired	43.0%	19.5%	34.4%	3.1%	
Other conditions	45.7%	15.5%	25.0%	13.8%	
In the city where you live, what is the approximate number of inhabitants?					
500.000 or more	37.3%	20.0%	24.0%	18.7%	0.015
250.000–499.999	43.7%	19.3%	21.0%	16.0%	
100.000–249.999	35.1%	18.9%	29.8%	16.2%	
20.000–99.999	40.3%	21.9%	22.7%	15.1%	
10.000–19.999	54.8%	16.5%	17.0%	11.7%	
Less than 10.000	42.6%	18.4%	24.6%	14.5%	
How many people currently live in the house where you are spending your social distancing period or most of your time?					
I live alone	39.2%	20.4%	28.4%	12.0%	0.249
2 persons	40.9%	21.2%	23.1%	14.7%	
3 persons	43.0%	18.1%	23.6%	15.4%	
4 or more persons	44.1%	18.1%	21.4%	16.4%	
How many children do you have living with you all or some of the time					
None	36.6%	21.3%	23.6%	18.6%	0.005
One	45.8%	17.8%	22.6%	13.9%	
Two	44.7%	18.0%	24.7%	12.6%	
Three or more	46.8%	19.0%	21.8%	12.4%	
Generally, how is your health?					0.000
Very good/good	22.2%	24.5%	14.8%	22.2%	
Nor good or bad	11.5%	20.8%	15.6%	11.5%	
Bad/very bad	14.0%	20.6%	15.0%	14.0%	
Are you suffering from chronic diseases or long-lasting health problems?					0.001
Yes	48.3%	17.6%	22.5%	11.6%	
No	39.3%	20.3%	24.0%	16.5%	
Due to health problems, do you have any limitations in your daily activities?					0.006
Serious limitations	50.0%	11.8%	24.1%	14.2%	
Not serious limitations	46.3%	15.8%	24.5%	13.4%	
No limitations	40.2%	21.2%	23.1%	15.5%	
Depression level					0.000
No depression	35.8%	28.4%	24.7%	11.1%	
Minimal	37.7%	24.1%	25.2%	13.0%	
Mild	43.3%	18.5%	21.7%	16.6%	
Moderate	44.7%	14.2%	22.8%	18.3%	
Moderate severe or severe	49.8%	11.0%	25.6%	13.7%	
Before social distancing, did you regularly practice sports?					0.000
Yes	22.2%	33.9%	41.0%	2.9%	
No	65.7%	2.3%	2.8%	29.2%	

Values in percentages. *p*: *p* of Chi-squared test.

**Table 3 ijerph-18-05070-t003:** Results of the multivariate regression for the PA practice during the period of social distancing (*n* = 2099, 387 males and 1712 females).

Variable	Practice with the Same Frequency as Before **		Practice, Although Less Frequently **		Yes, Started to Practice in Social Distancing Period **	
	**OR**	95% CI	**OR**	**95% CI**	**OR**	**95% CI**
Age category						
18–24	1.04	0.33–3.25	0.33 *	0.12–0.89	1.76	0.29–10.55
25–34	0.96	0.35–2.66	0.32 *	0.14–0.75	1.36	0.24–7.68
35–44	0.72	0.26–1.99	0.46 *	0.2–1.06	1.13	0.2–6.38
45–54	0.86	0.31–2.36	0.37 *	0.16–0.86	0.82	0.14–4.68
55–64	1.00	0.39–2.55	0.49	0.23–1.06	1.19	0.22–6.57
65+	1.00	-	1.00	-	1.00	-
Educational level						
Lower secondary school or less	0.35*	0.2–0.62	0.22	0.12–0.39	0.59	0.31–1.12
Diploma/upper secondary school	0.54*	0.34–0.88	0.47	0.3–0.75	0.80	0.47–1.37
Degree	0.80	0.55–1.17	0.95	0.66–1.37	1.07	0.69–1.68
Master, PhD or post-graduate degree	1.00	-	1.00	-	1.00	-
Professional status						
Worker	0.91	0.5–1.65	0.72	0.43–1.22	1.03	0.54–1.94
I am looking for a new job	0.88	0.42–1.84	0.49 *	0.25–0.98	0.96	0.44–2.08
Housewife	0.50	0.18–1.33	0.59	0.26–1.34	0.68	0.24–1.88
Student	0.83	0.37–1.86	0.70	0.33–1.47	1.38	0.62–3.06
Retired	0.71	0.28–1.79	0.72	0.33–1.61	0.20 *	0.05–0.92
Other conditions	1.00	-	1.00	-	1.00	-
In the city where you live, what is the approximate number of inhabitants?						
500.000 or more	1.08	0.7–1.67	1.04	0.69–1.56	1.25	0.79–1.96
250.000–499.999	1.00	0.57–1.75	0.76	0.44–1.32	0.93	0.51–1.7
100.000–249.999	1.23	0.79–1.92	1.51 *	1.02–2.24	1.31	0.83–2.09
20.000–99.999	1.16	0.84–1.6	0.86	0.63–1.17	0.95	0.66–1.36
10.000–19.999	0.62 *	0.39–0.98	0.51 *	0.34–0.78	0.59 *	0.36–0.97
Less than 10.000	0.00	-	0.00	-	0.00	-
Generally, how is your health?						
Very good/good	1.20	0.61–2.35	1.81	1–3.29	0.88	0.45–1.71
Nor good or bad	0.58	0.29–1.16	1.20	0.66–2.19	0.85	0.43–1.65
Bad/very bad	1.00	-	1.00	-	1.00	-
Are you suffering from chronic diseases or long-lasting health problems?						
Yes	1.16	0.84–1.61	0.84	0.62–1.14	0.72	0.5–1.04
No	0.00	-	0.00	-	0.00	-
Due to health problems, do you have any limitations in your daily activities?						
Serious limitations	0.60	0.35–1.01	1.21	0.79–1.85	1.03	0.63–1.68
Not serious limitations	0.78	0.54–1.14	1.16	0.83–1.62	0.97	0.65–1.46
No limitations	1.00	-	1.00	-	1.00	-
Depression level						
No depression	2.69 *	1.44–5.04	0.91	0.52–1.6	1.35	0.66–2.73
Minimal	2.00 *	1.2–3.35	0.86	0.57–1.31	1.23	0.74–2.06
Mild	1.51	0.9–2.51	0.75	0.5–1.12	1.39	0.86–2.26
Moderate	1.22	0.69–2.14	0.85	0.54–1.32	1.42	0.85–2.39
Moderate severe or severe	1.00	-	1.00	-	1.00	-

*: *p* < 0.05; ** Note: Reference category: No practice sports during the period of social distancing.

**Table 4 ijerph-18-05070-t004:** Results of the multivariate regression for the PA practice changes during the period of social distancing (*n* = 2099, 387 males and 1712 females).

Variable	FROM YES TO NO **		FROM NO TO YES ***	
	OR	95% CI	OR	95% CI
Age category				
18–24	0.34	0.1–1.21	1.71	0.38–7.65
25–34	0.56	0.19–1.65	0.96	0.23–3.95
35–44	0.58	0.2–1.71	0.88	0.21–3.63
45–54	0.43	0.15–1.27	0.70	0.17–2.89
55–64	0.50	0.2–1.29	0.92	0.23–3.63
65+	1	-	1	-
Educational level				
Lower secondary school or less	0.51 *	0.26–1	0.32 *	0.16–0.62
Diploma/upper secondary school	1.01	0.56–1.84	0.45 *	0.26–0.79
Degree	1.07	0.67–1.71	0.78	0.49–1.25
Master, PhD or post-degree	1	-	1.00	-
Professional status				
Worker	0.72	0.34–1.53	0.86	0.46–1.59
I am looking for a new job	1.23	0.43–3.53	0.81	0.38–1.72
Housewife	0.34 *	0.12–0.97	0.83	0.31–2.23
Student	0.92	0.33–2.57	1.00	0.45–2.22
Retired	0.35 *	0.13–0.96	0.45	0.12–1.65
Other conditions	1	-	1	-
In the city where you live, what is the approximate number of inhabitants?				
500.000 or more	0.97	0.58–1.62	1.16	0.71–1.9
250.000–499.999	0.98	0.49–1.95	0.74	0.38–1.42
100.000–249.999	1.31	0.77–2.22	1.33	0.81–2.18
20.000–99.999	1.04	0.7–1.55	0.93	0.63–1.35
10.000–19.999	0.49 *	0.3–0.8	0.63	0.38–1.03
Less than 10.000	1	-	1	-
Generally, how is your health?				
Very good/good	1.15	0.52–2.55	0.86	0.44–1.68
Nor good or bad	0.73	0.33–1.6	0.83	0.42–1.64
Bad/very bad	1	-	1	-
Are you suffering from chronic diseases or long-lasting health problems?				
Yes	1.09	0.74–1.6	0.66 *	0.45–0.97
No	1	-	1	-
Due to health problems, do you have any limitations in your daily activities?				
Serious limitations	0.88	0.49–1.58	1.27	0.78–2.06
Not serious limitations	0.98	0.64–1.51	1.02	0.67–1.56
No limitations	1	-	1	-
Depression level				
No depression	1.54	0.75–3.16	1.41	0.67–2.99
Minimal	1.54	0.89–2.66	1.32	0.77–2.25
Mild	0.96	0.57–1.6	1.44	0.86–2.43
Moderate	0.92	0.52–1.63	1.44	0.82–2.51
Moderate severe or severe	1	-	1	-

*: *p* < 0.05; ** Reference category: Stopped doing sport during confinement.; *** Reference category: Continued without sports practice during confinement.

## Data Availability

Data available on demand

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
