# Peer review of "The Effect of Social Isolation on Physical Activity during the COVID-19 Pandemic in France"

_ijerph, 2021, doi:10.3390/ijerph18105070_

Round 1

Reviewer 1 Report

I really think it is a great work and you have collected interesting results.  I let you some few comments to add if it is possible.

This is an ambitious review that aims to assess the impact of the COVID-19 pandemic on physical activity in France and the association of physical activity with psychological health during the COVID-19 pandemic. However, it is a good article which describes step by step all the main points that mentions but it should deep a little more in the main points that they suggest.

  1. Line 39: “Loneliness and social isolation are usually associated with poor mental and physical health”. Give some brief reason about it because it is going to write about it.
  2. Lines 52-53: “PA promotion to prevent the pandemic spread of these diseases and to improve populations’ health has been for decades a core objective of health strategies and policies globally”. There are recent studies on the effect of similar pandemics on the population indicate that the factors that have contributed the most to reducing the psychological impact of isolation at home were to have received clear and consistent information.
  3. Lines 60-62: “Previous pandemic crises, as for example that caused by the severe acute respiratory syndrome (SARS), caused serious public health consequences, not only linked to the viral infection per se. Indirect impacts on communities’ health haven’t been assessed systematically”. It I right but you could mention for example there are data from studies conducted in previous epidemics in Middle East Respiratory Syndrome (MERS), influenza A virus subtype H1N1, Ebola, or initiation of COVID-19 show that quarantine as a protective measure has psychological effects on the population.
  4. Lines 75-76: “Starting from this background, in this paper we will focus on the changes in lifestyles of a sample of French people during the first lockdown, in the months between March and May 2020, with an emphasis on PA”. It could do a brief lockdown with other countries where they had a stronger lockdown.

The results and methodology are an excellent work, no comments to add.

  1. Lines 243-245: “Finally, a last interesting aspect related to healthy practices: we note an interesting continuity in the practice of PA among retirees who practiced even before the first lockdown. This is certainly an interesting indicator of a healthy lifestyle that tries to remain so even in situations of crisis”. It could add a little bit about that aspect. Therefore we suggest reading other articles which give details of it.

There is a substantial and rapidly growing body of literature addressing these aims and a review is timely and would be of interest to many readers. Below are some examples of relevant studies that are not included. Some have been published after the final search date and some appear to have been missed. I appreciate that this is a fast moving area, but there is far too much relevant literature that has not been included in this article for it to meaningfully contribute to the literature.

  1. Callow et al The Mental Health Benefits of Physical Activity in Older Adults Survive the COVID-19 Pandemic.
  2. Duncan et al Perceived change in physical activity levels and mental health during COVID-19: Findings among adult twin pairs
  3. Faulkner et al Physical activity, mental health and well-being of adults during early COVID-19 containment strategies: A multi-country cross-sectional analysis (16 Julio)
  4. Ingram et al Changes in Diet, Sleep, and Physical Activity Are Associated With Differences in Negative Mood During COVID-19 Lockdown
  5. Meyer et al Changes in physical activity and sedentary behaviour due to the COVID-19 outbreak and associations with mental health in 3,052 US adults
  6. Pieh et al The effect of age, gender, income, work, and physical activity on mental health during coronavirus disease (COVID-19) lockdown in Austria
  7. Schuch et al Associations of moderate to vigorous physical activity and sedentary behavior with depressive and anxiety symptoms in self-isolating people during the COVID-19 pandemic: A cross-sectional survey in Brazil

Other comments:

The results section should focus more on the three aims of this review and less on the study characteristics.

It seems inconsistent that in the results it is stated that just one study focused on Aim 1 and in the Aim 1 subsection of the Discussion six relevant studies are cited. The same appears to be true for Aim 2

Author Response

Dear  Reviewer,

Thank you for giving us the opportunity to submit a revised draft of the manuscript “The effect of social isolation on physical activity during the Covid-19 pandemic in France” for publication in the “Journal of Environmental and Public Health”.

We appreciate the time and effort that you and the other reviewers dedicated to providing feedback on our manuscript and are grateful for the insightful comments on and valuable improvements to our paper (here attached). We have incorporated most of the suggestions made by the reviewers. Please see below for a point-by-point response to your comments and concerns.

Kind regards

I really think it is a great work and you have collected interesting results.  I let you some few comments to add if it is possible.

Thank you for your comment

This is an ambitious review that aims to assess the impact of the COVID-19 pandemic on physical activity in France and the association of physical activity with psychological health during the COVID-19 pandemic. However, it is a good article which describes step by step all the main points that mentions but it should deep a little more in the main points that they suggest.

Line 39: “Loneliness and social isolation are usually associated with poor mental and physical health”. Give some brief reason about it because it is going to write about it.

This sentence is linked to the following one, which partly explains it with the example of sociability, and extends it with the example of sedentariness. In order to clarify this statement,  in the text we have tied them up better by adding "an example".

Lines 52-53: “PA promotion to prevent the pandemic spread of these diseases and to improve populations’ health has been for decades a core objective of health strategies and policies globally”. There are recent studies on the effect of similar pandemics on the population indicate that the factors that have contributed the most to reducing the psychological impact of isolation at home were to have received clear and consistent information.

This sentence actually refers to the pandemic spread of diseases related to physical inactivity and sedentary lifestyles. We have made the sentence clearer (underlined in yellow in the document)

Lines 60-62: “Previous pandemic crises, as for example that caused by the severe acute respiratory syndrome (SARS), caused serious public health consequences, not only linked to the viral infection per se. Indirect impacts on communities’ health haven’t been assessed systematically”. It I right but you could mention for example there are data from studies conducted in previous epidemics in Middle East Respiratory Syndrome (MERS), influenza A virus subtype H1N1, Ebola, or initiation of COVID-19 show that quarantine as a protective measure has psychological effects on the population.

We appreciate the reviewer’s insightful suggestion.  We have supplemented a sentence that you can find underlined in yellow in the new text with a new reference too, concerning early studies on the psychological impact of lockdown and quarantine.

Lines 75-76: “Starting from this background, in this paper we will focus on the changes in lifestyles of a sample of French people during the first lockdown, in the months between March and May 2020, with an emphasis on PA”. It could do a brief lockdown with other countries where they had a stronger lockdown.

We have modified the last part of the introduction by including also a reference to the Chinese case, which was the centre of the epidemic when the spread of the virus was first detected and the lockdown was stronger.

The results and methodology are an excellent work, no comments to add.

Thank you!

Lines 243-245: “Finally, a last interesting aspect related to healthy practices: we note an interesting continuity in the practice of PA among retirees who practiced even before the first lockdown. This is certainly an interesting indicator of a healthy lifestyle that tries to remain so even in situations of crisis”. It could add a little bit about that aspect. Therefore we suggest reading other articles which give details of it.

There is a substantial and rapidly growing body of literature addressing these aims and a review is timely and would be of interest to many readers. Below are some examples of relevant studies that are not included. Some have been published after the final search date and some appear to have been missed. I appreciate that this is a fast moving area, but there is far too much relevant literature that has not been included in this article for it to meaningfully contribute to the literature.

Callow et al The Mental Health Benefits of Physical Activity in Older Adults Survive the COVID-19 Pandemic.

Duncan et al Perceived change in physical activity levels and mental health during COVID-19: Findings among adult twin pairs

Faulkner et al Physical activity, mental health and well-being of adults during early COVID-19 containment strategies: A multi-country cross-sectional analysis (16 Julio)

Ingram et al Changes in Diet, Sleep, and Physical Activity Are Associated With Differences in Negative Mood During COVID-19 Lockdown

Meyer et al Changes in physical activity and sedentary behaviour due to the COVID-19 outbreak and associations with mental health in 3,052 US adults

Pieh et al The effect of age, gender, income, work, and physical activity on mental health during coronavirus disease (COVID-19) lockdown in Austria

Schuch et al Associations of moderate to vigorous physical activity and sedentary behavior with depressive and anxiety symptoms in self-isolating people during the COVID-19 pandemic: A cross-sectional survey in Brazil

Thank you very much for these interesting tips. We have enlarged this paragraph by including two of the suggested references (Faulkner et al. and Callow et al.). Unfortunately, inserting the others would have made the "discussion" a bit scattered, but we think that those inserted enrich it a lot.

Other comments:

The results section should focus more on the three aims of this review and less on the study characteristics.

It seems inconsistent that in the results it is stated that just one study focused on Aim 1 and in the Aim 1 subsection of the Discussion six relevant studies are cited. The same appears to be true for Aim 2

For what concerns aims of the study, we clarified our hypothesis at the end of § 1 and in the discussion we tried to comment our findings considering other studies.  The six studies you mentioned regards prior pandemics in order to make a comparison.  We are not completely sure that what has been added can respond to requests: if it doesn’t, we kindly ask you to detail this point.

Reviewer 2 Report

  • Put your actions in past tense: Line 37: you say “…we will analyzed…” Change it to “…we analyzed…
  • Include a hypothesis at the end of your introduction
  • You refer to your population as “a sample of French people.” Can you make it more specific than that?

Methods

  • Was there inclusion and exclusion criteria for your study?
  • Please include the types of data that were collected (what types of questions, surveys, etc) and provide a brief description of each survey. You have a lot of data presented in your results, so it would be really helpful to describe the surveys that all the data are coming from.

Discussion

  • Please provide a little more detail to explain your results. For example, in the paragraph starting at line 211—Is there more research to support these findings besides one study?
  • For your limitations, it may also be important to note that those who are more active on social media during the pandemic and were willing to participate in the survey may have been more likely to increase PA compared to those who did not use social media

Author Response

Dear  Reviewer,

Thank you for giving us the opportunity to submit a revised draft of the manuscript “The effect of social isolation on physical activity during the Covid-19 pandemic in France” for publication in the “Journal of Environmental and Public Health”.

We appreciate the time and effort that you and the other reviewers dedicated to providing feedback on our manuscript and are grateful for the insightful comments on and valuable improvements to our paper (here attached). We have incorporated most of the suggestions made by the reviewers. Please see below for a point-by-point response to your comments and concerns.

Kind regards

Put your actions in past tense: Line 37: you say “…we will analyzed…” Change it to “…we analyzed…

We transformed our actions in the introduction by putting them in the present time.

Include a hypothesis at the end of your introduction

Thank you for your suggestion. At the end of the introduction, we included these hypotheses: (1) the pandemic had an impact on the frequency of physical activity; (2) socio-demographic characteristics and health conditions influence the likelihood of physical activity or not during the breakdown.

You refer to your population as “a sample of French people.” Can you make it more specific than that?

The characteristics of the sample are illustrated in the "Results" section. As for the "introduction" we changed the first sentence with “The objective of this cross-sectional study is to analyze the changes in physical activity’s (PA) practice of a sample of 2099 French adults, mostly females, who answered an online questionnaire during the first lockdown (March-May 2020)”.

Methods

Was there inclusion and exclusion criteria for your study?

Please include the types of data that were collected (what types of questions, surveys, etc) and provide a brief description of each survey. You have a lot of data presented in your results, so it would be really helpful to describe the surveys that all the data are coming from.

The "Methodology" section has been supplemented with a more complete illustration of the methodology (types of data, etc.) including the inclusion criterion. You can find it underlined in yellow in the point 2.1 “Study design” section

Discussion

Please provide a little more detail to explain your results. For example, in the paragraph starting at line 211—Is there more research to support these findings besides one study?

For your limitations, it may also be important to note that those who are more active on social media during the pandemic and were willing to participate in the survey may have been more likely to increase PA compared to those who did not use social media

We have strengthened the paragraphs that began previously on lines 207 and 211 by integrating additional bibliographical references and expanding a little bit the argument.

As for limitations, we found no data to support the hypothesis that those who are more active on social media during the pandemic and were willing to participate in the survey may have been more likely to increase PA compared to those who did not use social media. But we stressed the limits of our sampling strategy including this sentence at the beginning of the “limits” section: “The respondents were recruited through a "snowball" non-probabilistic sampling strategy, through the Facebook platform. This means that the people who responded to our questionnaire were also the most active on social media, especially on Facebook”. At the end of the “limits” section we have added the following sentence: “Additionally, the cross-sectional design limits the ability to draw on causal associations”.

Reviewer 3 Report

Thank you for the opportunity to read this very interesting paper. The paper has many strengths that should be of interest to the journal audience. Thus, the following suggestions are around enhancing the presentation for publication and clarifying aspects of the data and reporting.

I will go by line number for the most part. If not, I will try to be as specific as possible in noting the area I am speaking about.

Abstract

Authors should to include a little background of the study.

[Line 17] It is not clear the main aim. I recommend the structure: The objective of the study is to analyse… General French population?

When you say: “…during the first lockdown”. When was the first lockdown in France?

Authors must specify the type of study design. A cross-sectional study was carried out with a sample… Authors should speak about the sample before than statistical analysis.

[Line 25] … the probability (How much?) of starting to practice is greater…  Where are p-values? (p < .001). Authors must specify it.

  1. Introduction

[Line 37, page 1] Authors must finish 1. Introduction with the main aim.

  1. Materials and Methods

How was the sample chosen? Authors must specify it.

Do the authors have a study protocol? The study protocol should be described in detail.

It is necessary to include information about Design, Procedure and Sample (inclusion and exclusion criteria), Measuring Instruments, Ethical Considerations, etc.

[Line 103, page 3] What is it considering “incomplete questionnaires”? It is not clear.

Which is the ID number? (ID number…..:2020). Interventionary studies involving animals or humans, and other studies require ethical approval must list the authority that provided approval and the corresponding ethical approval code. Please include the date and code register number of ethics committee.

Are the Measuring Instruments adapted to the French population? Authors must justify their response. In this case more details are needed.

  1. Results

Table 1 and 2: please indicate where the analysis are statistically significant and where not

  1. Discussion

How these results can be globally interpreted?

  1. Conclusion

In my opinion, I think that Conclusion is very extensive.

[Lines 249-251] Please consider to move this sentence to the more appropriate Discussion section.

Limitations related with the type of methodology used. Authors must specify it.

References

Authors must review all references

  1. Author 1, A.B.; Author 2, C.D. Title of the article. Abbreviated Journal Name Year, Volume, page range.

I wish you all the best.

Author Response

Dear  Reviewer,

Thank you for giving us the opportunity to submit a revised draft of the manuscript “The effect of social isolation on physical activity during the Covid-19 pandemic in France” for publication in the “Journal of Environmental and Public Health”.

We appreciate the time and effort that you and the other reviewers dedicated to providing feedback on our manuscript and are grateful for the insightful comments on and valuable improvements to our paper (here attached). We have incorporated most of the suggestions made by the reviewers. Please see below for a point-by-point response to your comments and concerns.

Kind regards

Thank you for the opportunity to read this very interesting paper. The paper has many strengths that should be of interest to the journal audience. Thus, the following suggestions are around enhancing the presentation for publication and clarifying aspects of the data and reporting.

Thank you for your comment and for accepting to review our paper

I will go by line number for the most part. If not, I will try to be as specific as possible in noting the area I am speaking about.

Abstract

Authors should to include a little background of the study.

[Line 17] It is not clear the main aim. I recommend the structure: The objective of the study is to analyse… General French population?

When you say: “…during the first lockdown”. When was the first lockdown in France?

Authors must specify the type of study design. A cross-sectional study was carried out with a sample… Authors should speak about the sample before than statistical analysis.

We changed the first sentence: “The objective of this cross-sectional study is to analyze the changes in physical activity’s (PA) practice of a sample of 2099 French adults, mostly females, who answered an online questionnaire during the first lockdown (March-May 2020)”.

[Line 25] … the probability (How much?) of starting to practice is greater…  Where are p-values? (p < .001). Authors must specify it.

We have not included these elements because unfortunately the length of the abstract is limited to 200 words. We preferred to opt for a simple type of abstract without this information, which can be easily found in the "results" section and in the relevant tables. If the journal allows us to exceed the 200-word limit, we can add these elements to the abstract

Introduction

[Line 37, page 1] Authors must finish 1. Introduction with the main aim.

As suggested by another reviewer, hypotheses were included at the end of the introduction.

Materials and Methods

How was the sample chosen? Authors must specify it.

Do the authors have a study protocol? The study protocol should be described in detail.

It is necessary to include information about Design, Procedure and Sample (inclusion and exclusion criteria), Measuring Instruments, Ethical Considerations, etc.

Section 2.1 on "Study design" was implemented with elements concerning the study protocol, sampling, questionnaires used etc.

[Line 103, page 3] What is it considering “incomplete questionnaires”? It is not clear.

We have changed the text on line 137-138 (new version of the paper) with : “After excluding people who did not answer all questions”, that we think is more clear.

Which is the ID number? (ID number…..:2020). Interventionary studies involving animals or humans, and other studies require ethical approval must list the authority that provided approval and the corresponding ethical approval code. Please include the date and code register number of ethics committee.

The study was approved by the Ethics Committee of the Policlinico Gemelli, Catholic University of the Sacred Heart of Rome (Prot. N. 00255223/20). This has been now specified at the end of the section “2.1 Study design”

Are the Measuring Instruments adapted to the French population? Authors must justify their response. In this case more details are needed.

The measuring instruments are adapted to the French population, and allow comparison with other populations. Except for the PHQ-8 for which a validated translation into French already existed, the translation of the questionnaires was carried out according to the classical procedures of intercultural translation (Chapman & Carte, 1979; Cha et al., 2007). First, the questionnaire was translated from Italian to French using the parallel back-translation procedure (Brislin, 1986), in which a bilingual person translates the questionnaire from his or her original language into the language under study. Another bilingual person, who was not familiar with the original questionnaire, back-translated this version into the original language to assess discrepancies etc.

If you think that this protocol should be incorporated into the paper, it will be.

Results

Table 1 and 2: please indicate where the analysis are statistically significant and where not

“The value of "p" is specified in the last column on the right-hand side of the tables.”?

Discussion

How these results can be globally interpreted?

As you suggested, we moved lines 249-251 in the discussion, we think that they provide an overall interpretation of the results.

Conclusion

In my opinion, I think that Conclusion is very extensive.

[Lines 249-251] Please consider to move this sentence to the more appropriate Discussion section.

We have moved this sentence at the end of the “discussion” section

Limitations related with the type of methodology used. Authors must specify it.

We have added the following sentence at the end of the “limits” section: “Additionally, the cross-sectional design limits the ability to draw on causal associations”.

References

Authors must review all references

Author 1, A.B.; Author 2, C.D. Title of the article. Abbreviated Journal Name Year, Volume, page range.

The bibliography has been updated and amended

I wish you all the best.

Thank you, fingers, crossed!

Reviewer 4 Report

Review of the article The effect of social isolation on physical activity during the Covid-19 pandemic in France

Thank you for inviting me to the review.

Abstact

There is no information about the methods and tools used for PA analysis Introduction Lack of basic information about the Covid 19 pandemic, when and where did it start, what were the consequences and losses?

Materials and Methods

The authors write that the research results are part of the survey report in several European countries, but there is no basic information - What was the sample selection, what research tools and methods were used, etc.

Results -

All tables contain information on the level of depression in the opinion of the respondents, which the authors received on the basis of the PHQ-9 questionnaire. On what basis did the authors present the results of the research divided into five categories? (Non depressiona, minimal. Mild…). There are 9 questions in total in the PHQ-9 questionnaire, and the respondent marks the answers on a scale from 0 to 3. The higher the result, the greater the severity of depression, etc.

Discussion

The authors write that they used the PHQ-9 questionnaire - is it a full questionnaire or a short version, is the questionnaire validated in France? There is no information in the discussion about the results from the report from other countries (Italy, Great Britain, Sweden, Czech Republic, Poland). This information can be compared with the results and complement the discussions.

Conclusions - ok 

Author Response

Dear  Reviewer,

Thank you for giving us the opportunity to submit a revised draft of the manuscript “The effect of social isolation on physical activity during the Covid-19 pandemic in France” for publication in the “Journal of Environmental and Public Health”.

We appreciate the time and effort that you and the other reviewers dedicated to providing feedback on our manuscript and are grateful for the insightful comments on and valuable improvements to our paper (here attached). We have incorporated most of the suggestions made by the reviewers. Please see below for a point-by-point response to your comments and concerns.

Kind regards

Review of the article The effect of social isolation on physical activity during the Covid-19 pandemic in France

Thank you for inviting me to the review.

Thank you for accepting to review it

Abstact

There is no information about the methods and tools used for PA analysis

We have not included these elements in the abstract because unfortunately its length is limited to 200 words. We preferred to opt for a simple type of abstract without this information, which can be easily found in the "results'' section and in the tables. As you can see, we have modified the beginning of the asbtract a little to make it clearer, but unfortunately further modifications would make it too long and therefore not suitable for the magazine. However, if the journal allows us to exceed the 200-word limit, we can add these elements to the abstract

Introduction Lack of basic information about the Covid 19 pandemic, when and where did it start, what were the consequences and losses?

An introductory paragraph providing basic information on the emergence of the pandemic and the situation in France during the first wave has been included.

Materials and Methods

The authors write that the research results are part of the survey report in several European countries, but there is no basic information - What was the sample selection, what research tools and methods were used, etc.

The "Methodology" section has been supplemented with a more complete illustration of the methodology (types of data, tools, etc.) including sample selection and inclusion criterion. You can find it underlined in yellow in the point 2.1 “Study design” section.

Results -

All tables contain information on the level of depression in the opinion of the respondents, which the authors received on the basis of the PHQ-9 questionnaire. On what basis did the authors present the results of the research divided into five categories? (Non depressiona, minimal. Mild…). There are 9 questions in total in the PHQ-9 questionnaire, and the respondent marks the answers on a scale from 0 to 3. The higher the result, the greater the severity of depression, etc.

We used the PHQ-8, not the PHQ-9. Unfortunately, we made a mistake in the paper and listed the PHQ-9, but we have now corrected it. The PHQ-8 includes eight of the nine criteria on which the DSM-IV diagnosis of depressive disorders is based. The ninth question in the DSM-IV assesses suicidal or self-injurious thoughts and we excluded it as suggested in other research (because it is uncommon in the general population). We based the scoring on “Kroenke K, Strine TW, Spitzer RL et al. (2009) The PHQ-8 as a measure of current depression in the general population. J Affect Disord 114(1-3):163-173”

Discussion

The authors write that they used the PHQ-9 questionnaire - is it a full questionnaire or a short version, is the questionnaire validated in France? There is no information in the discussion about the results from the report from other countries (Italy, Great Britain, Sweden, Czech Republic, Poland). This information can be compared with the results and complement the discussions.

We used the french validated version of PHQ-8, available here: http://www.phqscreeners.com. Analyses for the other countries in the research group have not been carried out yet. Since this paper mainly focuses on PA, these analyses will be the subject of a forthcoming comparative paper.

Conclusions - ok

Thank you

Round 2

Reviewer 3 Report

All recommendations have been incorporated. I have no further comments.

Kind regards